# Afterimages: Their Neural Substrates and Their Role as Short-Term Memory in the Neural Computation of Human Vision

## Abstract

Afterimages are seemingly simple yet very intriguing visual phenomena. Presently, essentially all the textbooks in vision science and in perceptual psychology introduce these phenomena; meanwhile they also ubiquitously subscribe to an incorrect view that afterimages are due to some peripheral adaptation mechanisms occurring in the retina of the eye. The contrasting view is that afterimages originate in the brain: This view is not new at all, but only recently there has been accumulating a multitude of evidence pointing to its truthfulness. Two recent and critical lines of advances related to afterimages in vision science are as follows: 1. LeVay et al. (1985) discovered a representation of the physiological blind spot in Layer 4 of the cortical area V1 (hereafter, V1-L4) in the macaque monkey's brain, and Adams et al. (2007) discovered the same in the human brain; 2. Wu (2024) re-discovered the phenomenon of an observer seeing their own blind spot as an afterimage and correlated this phenomenon to the above neuroanatomical findings. Together, these advances essentially pinpoint the first-stage neural substrate for afterimages to V1-L4. Here we build upon these advances and establish a neural theory of afterimages consisting of the following tenets: 1. Positive and negative afterimages share the same neural substrate; 2. Afterimages should be viewed as short-term memory (STM) in the brain instead of as peripheral adaptation in the retina; 3. In terms of the neural computational architecture of any cortical area, STM is sandwiched between a feedforward neural network and a feedback counterpart—it may play a computational role for variable binding. Finally, we discuss potentially fruitful bi-directional interactions between perceptual & neuroscientific researches in biological vision on the one hand and computational & engineering endeavors on artificial vision on the other.

## 1 Introduction

Under certain visual conditions, when a viewer sees a stimulus, they may continue to see an image of the stimulus even after the physical stimulus has already disappeared: This visual phenomenon is known as an afterimage. A basic and prominent issue pertinent to afterimages is where they occur in the human visual system: Are they merely some adaptation mechanisms happening in the retina of the eye (hereafter, this view will be referred to as the Retinal View)? or are they a form of visual memory residing in the brain (hereafter, the Brain View)? Table 1 lists some major publications concerning afterimages since the time when Newton (1691) communicated his observation of afterimages: As we can see, there have been proponents of both of these two views. In the present paper, we will demonstrate that the Retinal View is erroneous and only the Brain View is correct.

Submitted to 39th Conference on Neural Information Processing Systems (NeurIPS 2025). Do not distribute.

Table 1: Major publications on the Retinal View vs. the Brain View with regard to afterimages

| Investigators | Phenomena / Arguments | The Retinal View | The Brain View |
|---|---|---|---|
| Newton (1691) | He observed interocular transfer of afterimages and briefly suggested: "It [afterimage] seems rather to consist in a disposition of the sensorium [the part of the brain for sensation] to move the imagination strongly" (p. 154). | | ✓ |
| Darwin (1786) | Adaptation: analogy between the retina and the muscular system | ✓ | |
| Binet (1886) | Interocular transfer (pp. 43-45) | | ✓ |
| Delabarre (1889) | Interocular differences of seeing the same afterimage | ✓ | |
| Craik (1940) | Retinal anoxia ("blinding" an eye by finger-pressing it) disrupts afterimage. | ✓ | |
| Weiskrantz (1950) | Afterimage may be produced from imagination. | | ✓ |
| Urist (1958) | Pushing an eye's ball, the eye's scene view shifts in position but its afterimage stays. | | ✓ |
| Loomis (1972) | Afterimages from long-duration light stimulation with little bleaching are correlated with visual appearance. | | ✓ |
| LeVay et al. (1985) | Discovery of a cortical representation of the physiological blind spot in V1-L4 in the macaque monkey's brain | | |
| Shimojo et al. (2001) | Filling-in visual surface may generate afterimage. | | ✓ |
| Tsuchiya and Koch (2005) | Interocular influence on afterimage formation | | ✓ |
| Adams et al. (2007) | Discovery of a cortical representation of the physiological blind spot in V1-L4 in the human brain, providing the neuroanatomical basis for Wu (2024) | | |
| Shevell et al. (2008) | Interocular misbinding of color and form | | ✓ |
| Zaidi et al. (2012) | Physiological recordings in the macaque monkey's brain | ✓ | ✓ |
| Dong et al. (2017) | The Breese effect (Breese, 1899): binocular rivalry between two eyes' afterimages slower than that with physical stimuli. | ✓ | ✓ |
| Kronemer et al. (2024) | Shared mechanisms between afterimages and visual imagery | | ✓ |
| Wu (2024) | Re-discovery of the La Hire phenomenon (i.e., seeing the physiological blind spots as afterimages), pinpointing such afterimages to the neural substrate V1-L4. | | ✓ |
| Kittikiatkumjorn et al. (2025) | Afterimage color is factored by color constancy | | ✓ |

The topic of afterimages is universally taught in all the textbooks in vision science (e.g., Palmer, 1999, pp. 105 & 109) and in perceptual psychology (such a course may be known as "Sensation and Perception"; e.g., Wolfe et al., 2021, pp. 153-155). About a decade ago, essentially all such textbooks had subscribed only to the Retinal View. Presently, the situation is changing—for example, citing Zaidi et al. (2012), Wolfe et al. (2021) advocate a hybrid view as follows: "Adaptation occurs at multiple sites in the nervous system, though the primary generators are in the retina" (p. 155): The Retinal View is still a component in this hybrid view; hopefully, the Retinal View would become totally abandoned in another decade or so.

As shown in Table 1, the Brain View regarding afterimages' localization in the human visual system is not new at all: Newton (1691) was already suggesting it. However, only in the last several decades, there have been accumulating many lines of evidence in support of the Brain View. In this respect, two particularly relevant and critical findings are as follows: (1) LeVay et al. (1985) delineated a representation of the physiological blind spot in Layer 4 of the primary visual cortex (also known as the cortical area V1; hereafter, Layer 4 of V1 will be referred to as V1-L4) in the macaque monkey's brain, and Adams et al. (2007) found the same in the human brain; (2) Wu (2024) re-discovered the phenomenon of a human observer being able to see their own physiological blind spot as an afterimage and correlated this phenomenon to the neuroanatomical finding by VeLay et al. (1985) and Adams et al. (2007). Together, these recent advances decisively and precisely pinpoint the first-stage neural substrate of afterimages to V1-L4.

In this paper, we will build upon the above-mentioned recent advances and establish a neural theory of afterimages consisting of the following tenets: 1. Positive and negative afterimages share the same neural substrate: The first-stage is V1-L4, and the subsequent stages are the layer 4s in other visual cortical areas—in this respect, we will substantiate the Brain View about afterimages into a concrete form; 2. Afterimages constitute a form of short-term memory (STM) in the brain; 3. In terms of the neural computational architecture of the brain, for each cortical area, STM is sandwiched between a feedforward neural network and a feedback counterpart—it may play a computational role for variable binding. Finally, we discuss potentially fruitful bidirectional interactions between perceptual & neuroscientific researches in biological vision on the one hand and computer science & engineering endeavors in artificial / machine vision on the other.

## 2   Physiological blind spots, afterimages, and neural localization

In this section, we will expand on the conference presentation by Wu (2024) at the Annual Meeting of the Psychonomic Society last year. As a systematic explanation regarding the relation between physiological blind spots and afterimages, this section is necessary for a complete understanding of why it is possible to deterministically pinpoint the first-stage neural locus of afterimages to V1-L4. (We understand that Wu (2024) is only an abstract and that he has not yet published his conference presentation as a full paper, but here we do acknowledge that the basic idea in this section comes from him.)

### 2.1   The physiological blind spot in the eye

We have a physiological blind spot in each of our eyes: It corresponds to a port of the eye's retina (anatomically known as the "optic disk") where no photoreceptors (i.e., rods and cones) exist, where optic nerve fibers exit the eye, and where blood vessels enter and exit the eye (i.e., arteries entering and veins exiting the eye). This anatomical feature of the eye is clearly seen in Figure 1(a) which shows an image of a human eye's retina as seen by an ophthalmologist (i.e., eye doctor) when examining someone else's eye with some retina imaging device. The shortened term "blind spot" may mean various things in different contexts; hereafter, we will use it to refer specifically to the physiological blind spot in the eye.

The blind spot was discovered by the French scholar Edme Mariotte in the 1660s: It is certainly an amazing scientific discovery. Mariotte's method demonstrating the blind spot, however, is about how to map it within the viewer's visual field, not about how to (consciously) see it. Presently, all the textbooks in perceptual psychology, vision science, neuroscience, and ophthalmology, when mentioning about the blind spot, describe this method only (e.g., Wolfe et al., 2021, p. 40).

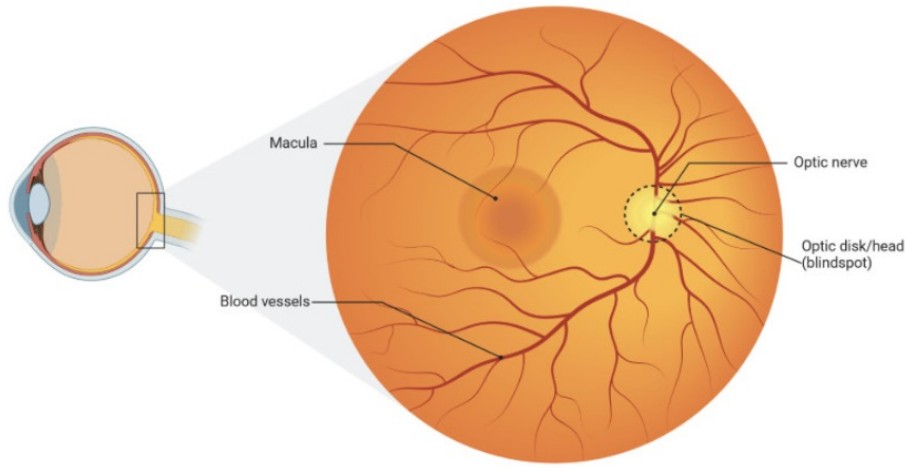

**(a) The retina of a human eye as seen by an ophthalmologist with an eye / retina imaging device. Optic Disk = Blind Spot (BS); the image of retinal blood vessels: Purkinje Tree (PT)**

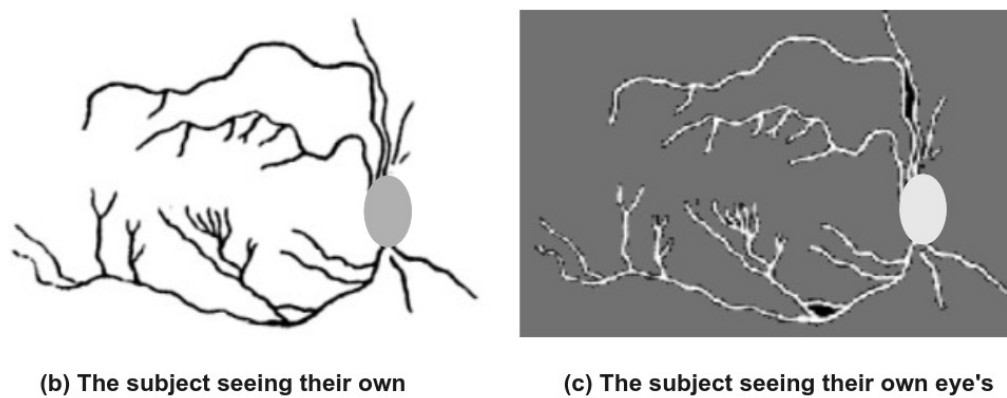

**(b) The subject seeing their own eye's BS & PT in positive form**

**(c) The subject seeing their own eye's BS & PT in negative form**

Figure 1: The blind spot and blood vessels within an eye as seen by ophthalmologist from outside and by the subject (the owner of the eye) within entopic vision.

Under special conditions, it is actually possible for the subject (i.e., the owner of the eye; we may also refer to him/her as the viewer or observer) to see their own blind spot in each eye, literally seeing the blind spot as a black hole on a lighter background or a white hole on a darker background, as illustrated in Figure 1(b) and © respectively—more generally, a colored spot on a background of the spot's complementary color; the BS may, or may not, be accompanied by the Purkinje Tree (PT) which denotes the image of retinal blood vessels. As far as we have been able to trace back, this phenomenon was first reported by the French scholar Philippe de La Hire (1640-1718) in La Hire (1694): Henceforth, we will refer to this phenomenon as the La Hire phenomenon. It was subsequently re-discovered by the Czech scientist Johann Evangelist Purkinje (1787–1869) in Purkinje (1819): Figure 1(b) is his drawing of his observation of his right eye's blind spot and retinal blood vessels. More broadly, Purkinje referred to a set of visual phenomena of a viewer seeing some characteristics of the human visual system's internal organization as "subjective vision"—presently, they are known as "entoptic vision"; therefore, the La Hire phenomenon is an instance of entoptic vision.

As mentioned by Helson (1929, pp. 352–353) and Brøns (1939, Chapter IV), many German psychologists had investigated the La Hire phenomenon before World War II. After the war, it appears that the vision research community has largely forgotten about this interesting visual phenomenon.

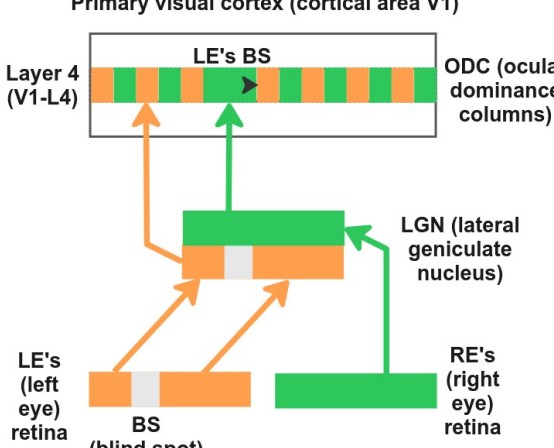

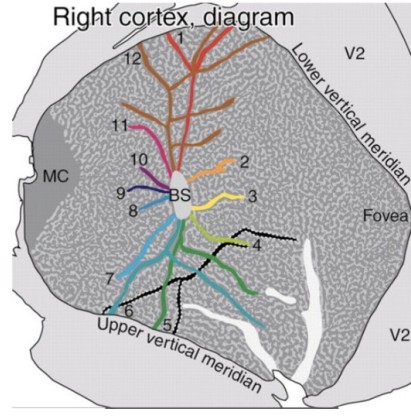

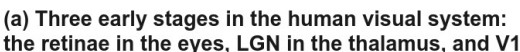

**(a) Three early stages in the human visual system: the retinae in the eyes, LGN in the thalamus, and V1**

**(b) Representations of the blind spot and retinal blood vessels in V1-L4**

Figure 2: Three early stages of the human visual system: retina, LGN, and V1-L4.

## 2.2 Seeing the blind spot as a negative afterimage

Wu (2024) described his observation of seeing an eye's blind spot as a negative afterimage—that is, normally, the blind spot would appear as a black spot on a neutral background; but under special conditions, as illustrated in Figure 1(c), it may appear negatively as a brighter spot on a darker background.

Actually, both Helson (1929) and Brøns (1939) had mentioned the possibility of seeing the blind spot as a negative afterimage: It is one of the characteristics of the La Hire phenomenon—in this respect, on the phenomenological side, Wu (2024) is a re-discovery; nonetheless, he correlated this phenomenological feature of the La Hire phenomenon and determined the neural locus of afterimages: We will describe this correlational reasoning below.

## 2.3 Localizing the neural substrate of the blind spot and its afterimage

Figure 2(a) illustrates the early stages of the human (or more broadly, the primate) visual system consisting of the retina, the lateral geniculate nucleus (LGN) of the thalamus, and the primary visual cortex (i.e., V1). The cortical sheet comprises six layers, with Layer 4 receiving thalamic inputs (i.e., optic radiations in the case of V1). Hereafter, we will denote this layer as V1-L4. Please note that "Layer 4" in V1 has been incorrectly labeled as "Layer 4C" in many textbooks (e.g., Wolfe et al., 2021, p. 71); see Boyd et al. (2000) and Balaram et al. (2014) for the relevant neuroanatomical evidence as to why it should be labeled as "Layer 4" instead of "Layer 4C".

The La Hire phenomenon is a wonderful psycho-anatomical means: As a matter of fact, several neuroanatomical studies have precisely localized a representation of the blind spot in V1-L4: LeVay et al. (1985) and Adams et al. (2007) (from Prof. Jonathan Horton's lab at University of California, San Francisco) are the two milestone discoveries in this regard, with the first one in the macaque monkey's brain and the second in the human brain. Though in two species and using different chemical staining methods, their central findings are essentially the same: There is a representation of the blind spot in V1-L4. For better illustration, Figure 2(b) shows a diagram of V1-L4 from a monkey's brain studied by Prof. Horton's lab. Please note that V1-L4 is a "bi-monocular" structure in the sense that for each and every tiny patch of the viewer's binocular visual field, the monocular image (i.e., ocular dominance column or ODC) from one eye resides, side by side, with that for the other eye. In Figure 2(b), white stripes and areas depict neural tissue regions in V1-L4 predominantly connected with the eye containing the blind spot, whereas the black stripes and areas depict that connected with the other. From this diagram, we should understand that the representation of the

blind spot in V1-L4 does not create any physical "hole" in this neural tissue—instead, the area is invaded and occupied by the input from the other eye.

Beyond V1-L4, is there any other neural structure(s) in the primate visual system that may contain representations of the blind spot? David Hubel and Torsten Wiesel's pioneering exploration of the feline and the primate visual brains had long established that neurons in V1-L4 are primarily monocular whereas that beyond V1-L4 are mainly binocular (e.g., Hubel & Wiesel, 1968). As we already stated, each eye's blind spot is specific to that eye (i.e., monocular); therefore, the answer to this question is negative. Correlating the La Hire phenomenon with such neuroanatomical studies, we can conclude that visual sensation is represented in V1-L4. Please note that without knowing the La Hire phenomenon, we cannot argue that the blind spot representations seen in V1-L4, and this layer in general, are directly correlated with visual sensations and afterimages—in other words, one may argue that such representations are just for sub-consciousness neural activation. With the knowledge of the La Hire phenomenon, then, we can indeed pinpoint the neural substrate for visual sensations and afterimages to V1-L4. Please note that the authors of the relevant neuroanatomical studies did not link their findings with any visual phenomenon; the correlation between the afterimage phenomenon and neuroanatomical underpinnings was advanced by Wu (2024).

# 3  Positive and negative afterimages

## 3.1  The Franklin effect

One conceptual blocker to thinking of afterimages as a form of memory is that an afterimage can manifest itself as a positive or negative one. Presently, the prevailing conception about positive and negative afterimages is that they are due to different physiological causes. For instance, De Valois and De Valois (1997) suggested that positive afterimages is due to temporary persistence of discharges of ganglion neurons in the retina, whereas negative afterimages is partly due to retinal photopigment bleaching and partly due to neural adaptation at an opponent-colors stage. Likewise, Gregory (2004, p.15) presents essentially the same conception about positive and negative afterimages. The conception that positive and negative afterimages are due to separate physiological processes (or mechanisms) is schematically illustrated in Figure 3(a).

This above conception, however, is just a misconception—this is because positive and negative afterimages can be mutually converted into one another; more specifically, an afterimage can appear either positive or negative depending on whether the observer is viewing it with his/her eyes closed or open; and with the eyes open, depending on whether projecting the afterimage onto a dark, gray, or white background. This phenomenon was first observed and described by Benjamin Franklin (1706—1790) [whose life was simply too many-splendored: a founding father of the United States, a successful businessman, a scientist famous for taking electricity from the sky, and an inventor]. On June 2, 1765, in a letter to Lord Kames, Franklin described his following observation: "A remarkable circumstance attending this experiment, is, that the impression of forms is better retained than that of colors; for after the eyes are shut, when you first discern the image of the window, the panes appear dark, and the cross bars of the sashes, with the window frames and walls, appear white or bright; but, if you still add to the darkness in the eyes by covering them with your hand, the reverse instantly takes place, the panes appear luminous and the cross bars dark. And by removing the hand they are again reversed." (Franklin, 1765, p.380). This phenomenon had been further studied by Robert Waring Darwin (1766—1848), the father of the evolutionist Charles Darwin (1809—1882). On March 23, 1786, Darwin read a paper on "ocular spectra" (that was the term for afterimages at that time) before the Royal Society of London; the paper was subsequently published as Darwin (1786).

The above phenomenon has been named the Franklin effect (Roeckelein, 2006, p. 649)—but to a large extent, somehow unfortunately, it remains largely unknown besides being mentioned in some comprehensively and meticulously compiled works, such as those by Roeckelein. As illustrated in Figure 3(a), the hypothesis that positive and negative afterimages are due to different physiological processes cannot account for the Franklin effect; therefore, we would need to seek other explanations for positive and negative afterimages.

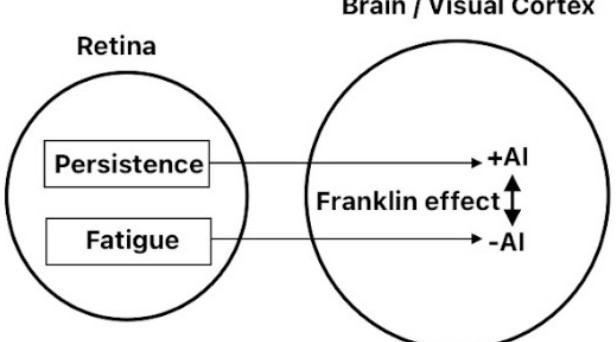

(a) Currently prevailing view:
+AI and -AI have separate
physiological processes in the retina
(e.g., Gregory, 2004, p.15);
however, this view cannot
explain the Franklin effect.

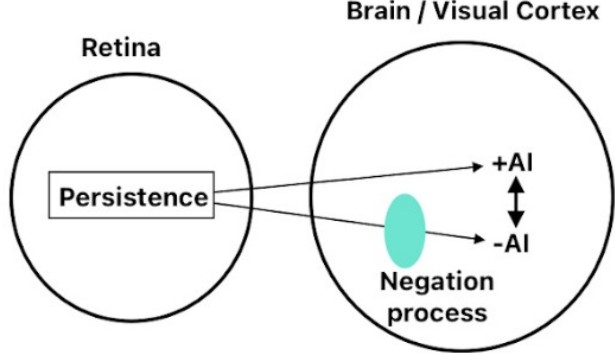

(b) McDougall's View:
+AI and -AI share the same
persistence process in the retina
(McDougall, 1901b, p.365)

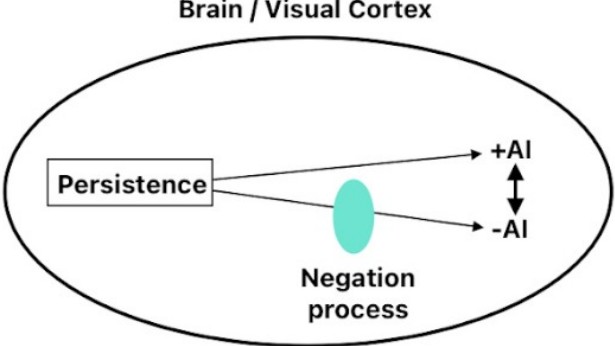

(c) The view advanced in this paper:
+AI and -AI share the same
persistence / memory process
in the brain / visual cortex.

Figure 3: Three views regarding positive and negative afterimages

### 3.2 McDougall's view regarding positive and negative afterimages

One and a quarter centuries after Franklin and Darwin, McDougall (1901a, 1901b) rediscovered the Frank effect, and then he further claimed: "...all afterimages, negative and positive, same-colored and complementary-colored alike, are primarily due to the persistence in the retina of X-substances ..." (McDougall, 1901b, p.365). As illustrated in Figure 3(b), his claim consists of two parts: (1) positive and negative afterimages are both due to some material persistence in our visual system; (2) this persistence resides in the retina of the eye. In the previous section, however, we have already dismissed afterimage's retinal origin; therefore, we can now adopt McDougall's view about positive and negative afterimages and modify it to become the view illustrated in Figure 3(c).

## 4 Afterimages as visual short-term memory (STM)

Now we have established two facts: afterimage is cortical in origin, and positive and negative afterimages are both due to neural persistence. In our opinion, a neural persistence process (or mechanism) in the brain should better be conceived as visual STM—as a matter of fact, an elementary form of this view had already been suggested by Newton in 1704.

During the years 1661—1664, when Newton was an undergraduate student at Trinity College, he kept a notebook which has passed down in history and is currently in archive at University of Cambridge Library (see McGuire & Tamny, 1983). The notebook contains a section under the heading "some philosophical questions"—there, Newton wrote down a wide range of observations and questions in natural philosophy, some of them belonging to perceptual and cognitive psychology as we know today, within the topics ranging from vision, audition, memory, imagery, to consciousness. For vision, he recorded a number of visual phenomena: One of them is a particular form of positive afterimages, as he described in this way: "There is required some permanency in the object to perfect vision, thus a coale whirled round is not like a coale but fiery circle ..." (McGuire & Tamny, 1983, p. 387). About 40 years later, when Newton (1704) published his book "Opticks", he did compile this visual phenomenon as one of the queries appended near the end of the book: "Query 16: ... And when a Coal of Fire moved nimbly in the circumference of a Circle, make the whole circumference appear like a Circle of Fire; is it not because the Motion excited in the bottom of the Eye by the Rays of Light are of a lasting nature, and continue till the Coal of Fire in going round returns to its former place?" (see McGuire & Tamny, 1983, p. 237)

What Newton was describing as "permanency" in his Trinity College notebook and as "lasting nature" in his book "Optiks" happens within the observer's mind—apparently, it is a form of memory in the human brain. Furthermore, Newton did point out that this form of positive afterimages plays an active functional role in color perception—specifically, in temporal color summation (also known as color mixture or color fusion)—see "Persistence of Vision" (2025).

Now, once we understand that afterimages play an active computational role in human vision, should we conceptualize afterimages better as STM than merely as adaptation? We suggest and believe that the answer is "yes".

## 5 Interdisciplinary Interactions between Cognitive Science & Neuroscience and Computer Science & Engineering

If one learns about afterimages from an introductory textbook in vision science, one would easily get an impression that all about afterimages have already been researched and known. This is very far from the truth. In the present paper, we have witnessed that on the one hand there have been a vast array of observational and experimental data about afterimages accumulated over the last 350 years or so (after Newton's recording of his observations on afterimages around 1664), and on the other, some basic questions concerning afterimages have not yet been fully answered. We have summarized some recent advances and theorized some functional or computational aspects about afterimages, but our paper certainly has limitations: One of them is that we have not yet established a quantitative model encompassing all or most of the empirical data—in this regard, we believe that interactions between cognitive science & neuroscience on the one hand and computer science & engineering on the other would become fruitful—such interactions may eventually to solve some fundamental problems in neuroscience, including those basic questions surrounding afterimages.

As narrated by Sejnowski (2018), Geoffrey Hinton, the godfather of deep learning, was closely associated with cognitive science neuroscience at UCSD, CMU, and UCL before his distinguished tenure at Google. Sejnowski himself was also in the same kind of interdisciplinary research environment and was closely connected with Hinton (Sejnowski, 2018, p. 60); as a pioneer and prominent computational neuroscientist, Sejnowski's research interests and research certainly lean more on the neuroscience side: As he summons in his book, there is still a vast scientific landground about human perception, cognition, and intelligence yet to be explored.

# 6 Conclusions

Currently, the standard textbook teaching about afterimages is that they origin in the retina of the eye. Here we have presented an array of evidence—particularly the phenomenon of the blind spot visible as an afterimage—to argue for cortical origins of afterimages. Furthermore, we have developed a new theoretical perspective for understanding afterimages in human vision, consisting of the following tenets: 1. Positive and negative afterimages share the same neural substrate; 2. Afterimages should be viewed as short-term memory (STM) in the brain instead of as peripheral adaptation; 3. In terms of the neural computational architecture of the brain, this STM is sandwiched between a feedforward neural network and a feedback counterpart—it may play a role for variable binding.

NeurIPS is a great venue for interdisciplinary interactions between cognitive science & neuroscience on the one hand, and computer science & engineering on the other. Our paper belongs to the former—hopefully, it would attract some talents and efforts from the latter to tackle the scientific issues raised here in one direction and to transfer and incorporate some of our ideas proposed here into real-world applications in the other.

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
