# OpenReview forum: "Afterimages: Their neural substrates and their role as short-term memory in the human brain’s computation."
_NeurIPS.cc/2025/Conference — Submitted to NeurIPS 2025_

### Official Review · Reviewer_KPHd · 2025-06-02

**Clarity:** 2
**Significance:** 1
**Originality:** 2
**Rating:** 1
**Confidence:** 5

**Summary:**

This paper discusses the neural substrated of visual afterimages. It argues against the idea that visual afterimages are caused by adaptation of peripheral neurons (e.g. photoreceptors), but instead are of cortical origin. Here, visual afterimages are viewed as a originating between feedforward and feedback processing potentially in layer 4 of V1.

**Questions:**

What are the sources of the figures reused in the paper and is their use covered by the licences?
What follows from the discussion in this paper for computational theories of vision and how could this be formalized?

**Ethical Concerns:**

["Major Concern: Data privacy, copyright, and consent"]

**Final Justification:**

I maintain my view that this paper is not suitable for Neurips, as it simply provides a literature review together with some opinion/hypotheses parts. I also maintain that the use of figures published by others in this paper is questionable.

**Limitations:**

The authors did not address or discuss any limitations.

**Paper Formatting Concerns:**

The paper cites an abstract Chen (2024) which appears to be an early version of the paper. Could be a breach of anonymity.

**Quality:**

1

**Strengths And Weaknesses:**

Strengths:
- The paper introduces an extensive historical overview of afterimages and concise summary of literature findings in Table 1.

Weaknesses:
- The paper does not present any new data, new models, new algorithms, new mathematical theory or empirical evaluation. It is a literature discussion of different findings regarding afterimages.
- Figure 1a looks like it is reproduced from a textbook. What is the source? Also Fig. 2b is Fig. 10c from Adams and Horton 2003, JNeurosci, but no source is credited in the caption. Also reproduction of this figure is likely not covered by the licence of the original paper.
- The style of the paper is sometimes that of a review, sometimes of a newspaper article, and often colloquial and of "storrytelling" nature (e.g. lines 234 ff).

---

> ### Author Rebuttal · Authors · 2025-07-31
>
> Thank you very much for your careful review of our NeurIPS submission!
>
> ********************
> Weakness-1: “The paper does not ...”
>
> In our opinion, we have two original contributions in our paper:
>
> 1. Correlating the La Hire phenomenon on the perceptual side with the neuroanatomical findings about the blind spot’s representation in V1-L4 by LeVay et al. (1985) in the monkey brain and by Adams et al. (2007) in the human brain. Please note that linking two previously seemingly-separate phenomena or mechanisms is also an original contribution to scientific advances. Neither LeVay et al. (1895) nor Adams et al. (2007) had made this correlation. As we are the first to suggest this correlation, we do perceive our contribution in this regard as original.
>
> 2. After locating the neural substrate of visual afterimages to V1-L4, we have further proposed a computational architecture for V1 (the primary visual cortex): V1-L4 is the STM for afterimages, V1-L2&3 is the feedforward network, and V1-L4&5 is the feedback network. Please note that the cortical sheet of the human brain mainly consists of 6 layers, but L1 contains mostly neural fibers; only L2-L6 contains neurons. Our computational architecture accounts for these 5 neuron-containing layers. Of course, our current description of this proposed computational architecture needs to be elaborated. We will further address this matter in our answer to your Question-2 below.
>
> Related references here:
>
> LeVay, S., Connolly, M., Houde, J., & Van Essen, D. C. (1985). The complete pattern of ocular dominance
>
> Adams, D. L., Sincich, L. C., & Horton, J. C. (2007). Complete pattern of ocular dominance columns in human primary visual cortex. Journal of Neuroscience, 27, 10391–10403.
>
> ********************
> Weakness-2: “Figure 1a looks ...”
>
> Regarding Figure 1a, first of all, the knowledge illustrated by this figure is already a piece of well-established and shared scientific knowledge: The blind spot of the eye was discovered by the French scholar Edme Marriotte by 1686 (for the history about the blind spot, see Brøns, 1939); Mariotte himself did not provide any diagram to illustrate the eye’s anatomy with its blind spot; on the other hand, a textbook in ophthalmology (about human eyes) published in 1759 already contained this kind of diagram, in black-and-white, to illustrate the overall structure of the human eye’s retina (Porterfield, 1759, Vol. 1, Figure 4). In this sense, this piece of knowledge is already in the public domain. As to our colored Figure 1a in specific, we took it from a popular-science website because it is simple (just enough to illustrate the blind spot and the retinal blood vessels in the eye’s retina) and is in color. We will either cite the source website or draw a sketchy illustration by ourselves.
>
> Related references here:
>
> Brøns, J. (1939). The Blind Spot of Mariotte: Its Ordinary Imperceptibility or Filling-in and Its Facultative Visibility. H. K. Lewis & Co.
>
> Porterfield, W. (1759). A Treatise on the Eye: the Manner and Phenomena of Vision. Hamilton and J. Balfour at Edinburgh.
>
> Regarding Figure 2b, thank you very much for tracking down these references and the relevant figures from Prof. Jonathan Horton and his colleagues’ work.  Currently, our Figure 2b is indeed Figure 10c from Adams and Horton (2003)—which is a study with the monkey brain. As we actually intend to cite their work on the human brain, we will replace this figure with Figure 7B from Adams, Sincich, and Horton (2007)—which is a study with the human brain. Moreover, we will definitely add the source in the capture of the figure. The “Journal of Neuroscience” is published by the Society for Neuroscience, and one of us is a member of this society: If our paper is accepted by NeurIPS, we will certainly obtain the permission to use this figure from the Society for Neuroscience before we finalize our paper for publication.
>
> References:
>
> Adams, D. L., & Horton, J. C. (2003). The representation of retinal blood vessels in primate striate cortex. Journal of Neuroscience, 23, 5984-5997.
>
> Adams, D. L., Sincich, L. C., & Horton, J. C. (2007). Complete pattern of ocular dominance columns in human primary visual cortex. Journal of Neuroscience, 27, 10391–10403.
>
> ********************
> Weakness #3. “The style of the paper ...”
>
> Our paper is heavily oriented on the neuroscience and cognitive science sides; to introduce the recent relevant advances in vision research to computer scientists and engineers, the first part of our paper does have a review flavor.
>
> In the earlier years of NIPS/NeurIPS, papers on the neuroscience and cognitive science sides occupied a quite significant portion. Since the early 2010s, as the deep learning framework has taken a great leap and become very successful in a wide range of applications, NeuroIPS has been attended more and more prominently by computer scientists and engineers. Thankfully, by and large, NeurIPS has kept its interdisciplinary tradition and kept the “Neuroscience and Cognitive Science” track; this “storytelling” part is intended to reflect upon the interdisciplinary tradition of NeurIPS. Thanks to your comment here, we will revise this part accordingly.
>
> ********************
> Question 1: “What are the sources …?”
>
> We have addressed this matter in our answer to Weakness-2 in the above.
>
> ********************
> Question 2: “What follows …?”
>
> Thank you very much for your very insightful question! There are two perspectives to this question:
>
> 1. The perspective from neuroscience and cognitive science
>
> As described by the British psychologist McDougall in 1901, under careful observations, some aspects of visual afterimages are quite similar to some diffusion and hysteresis phenomena observed in chemical reactions; hence, he had suggested that a chemical substance in the human eye and/or brain is responsible for afterimages—as various types of neurotransmitters were unknown at his time, McDougall referred to this candidate neurotransmitter as “X-Substances” (McDougall, 1901, p. 80).
>
> We have replicated many of McDougall’s observations about afterimages, and therefore we believe that this idea about X-Substances is worthy further consideration and exploration. In 1950s, it was discovered that the neurotransmitter “Glutamate” is the principal excitatory neurotransmitter from thalamic (that is, from the LGN) neural fibers’ terminals to cortical neurons within V1-L4 (Watkins & Jane, 2006). Correlating McDougall’s idea about X-Substances for afterimages and the present knowledge about Glutamate, we have quite naturally suggested that Glutamate’s diffusion and hysteresis is very likely the neurochemical basis for visual afterimages. One of us has presented this idea at a recent Society for Neuroscience annual meeting. We did not include this idea in our NeurIPS submission because we thought that it was too specific on the neuroscience side; as per your comment here, if our paper gets accepted, it is certainly better to be more specific with our proposed model for afterimages / STM and to include a description of this idea.
>
> We are also in collaboration with some vision scientists / psychophysicists trying to design some perceptual experiments, based on McDougall’s work (which has largely remained in obscurity, very unfortunately!), to further explore some behavioral aspects of visual afterimages and visual STM.
>
> References:
>
> McDougall, W. (1901). Some new observations ... Mind, 10, 52–97.
>
> Watkins, J. C., & Jane, D. E. (2006). The glutamate story. British Journal of Pharmacology. 147, S100–S108.
>
> 2. The computational perspective
>
> In this regard, our proposed computational architecture for the primate (including the human species) visual cortex is more relevant here—that is, each cortical area consists of three parts:  feedforward-STM-feedback. As of now, our description of this proposed computational architecture for the visual cortex is indeed not very detailed. We will certainly strengthen our paper in this regard: to describe it in more detail, and to include an information-flow diagram and a high-level algorithmic description.
>
> Our paper is more oriented on the neuroscience and cognitive science aspects; but as NeurIPS has kept its interdisciplinary tradition and as the major portion of NeurIPS attendees are from the computational domains (that is, computer scientists and engineers working on artificial neural networks, machine learning, computer vision, etc.), we believe that NeurIPS is a great and wonderful meeting venue for us to present our proposed computational architecture: Given such an opportunity, we are confident that our close interdisciplinary interactions with top talents in the computational domains would become very fruitful. We further believe that such interactions would run in two directions: (1) computer scientists would help us to further consolidate and substantiate our computational architecture as a scientific theory / model in neuroscience and cognitive science; (2) some computer scientists and/or engineers may find our computational architecture to be useful for some real-world applications.

---

> > ### Comment · Reviewer_KPHd · 2025-08-01
> >
> > I thank the authors for their reviews. Unfortunately, it does little to convince me that the paper is suitable for NeurIPS. I think it is simply out of scope.

---

> > > ### Author Response · Authors · 2025-08-08
> > > **Thank you very much for your comments! And here are our replies!**
> > >
> > > Thank you very much for your comment! We have combined comments from several sources, and have put together our replies below.
> > >
> > > “While linking existing phenomena can be valuable, you provide no quantitative analysis, computational modeling, or statistical validation to support this connection.”
> > >
> > > Actually, during this year’s Vision Science Society annual meeting, we did have a demo session about the La Hire phenomenon: Over 100 attendees participated in our experiment, and we did collect some data at the time. We thought that such data (e.g., percentages of the participants seeing the phenomenon under various conditions). We thought that such detailed psychological data would be very interesting to vision researchers, but not probably so much to computer scientists—to them, knowing the very existence of this phenomenon might already be quite informative; but as per your comment, we certainly can throw in some experimental data in our paper.
> > >
> > > From your perspective, what kind of quantitative data should we furnish? We will greatly appreciate hearing any thoughtful comment in this respect!
> > >
> > > Please note that the La Hire phenomenon is not new, but it had been forgotten after WWII (due to the fact that the old research publications about this phenomenon are mainly in German)—we verified this point by both systematically looking through the relevant textbooks & journals and by asking veteran vision researchers. Just as an example, Christoph Koch is a well-known vision scientist but he explicitly denied the possibility of seeing one’s own blind spots: “Yet even if you close one eye, you still won't see a hole in your visual field” (Koch, 2004, p. 54)—This statement is wrong! Under special viewing conditions, you can indeed see one eye’s blind spot as a hole (see our paper, p. 4, Fig. 1 & lines 86–98). This is the La Hire phenomenon, and one of us re-discovered it—now, we have been resurrecting this phenomenon in the vision research community.
> > >
> > > As to predictions on the neuroscience side, when we talked to those involving fMRI on the human brain, they immediately understand what to test and what experiments to do—but unfortunately, the current fMRI technology does not have the spatial resolution to localize brain activity to a specific layer within the cortical sheet along the pia-to-white-matter axis. But fMRI technology is advancing: Siemens has already built ultrahigh resolution fMRI scanners with 11.7T magnetic field—currently, they are not safe for doing experiments with living human beings. We may see such experiments within a decade or so.
> > >
> > > Why do we think that “knowing the very existence of this phenomenon may already be quite informative”. As an example, Daniel Dennett was a well-known philosopher and his book “Consciousness Explained” contains lengthy discussions about the blind spot. Based on the assumption that it is impossible to see one’s own blind spots, he proposed his No-Representational Theory for the Blind Spot (that is, claiming that there is no neural representation for the blind spot at all in the whole brain). As we mentioned above, the La Hire phenomenon had been forgotten by the vision research community after WWII—apparently, Dennett did not know this phenomenon; had he known it, he certainly would not have proposed his incorrect theory at all. Therefore, armed with just this one bit of information (that is, knowing the existence of the La Hire phenomenon), philosophers and scientists alike will see a new perspective for further investigation—or say, for making more grounded bold guesses (as Newton once remarked: “No great scientific progress would be made without bold guesses!”).
> > >
> > > “The absence of retinal tissue at the blind spot doesn't prove V1-L4 is the afterimage substrate - it only shows cortical involvement is necessary for that specific location.”
> > >
> > > Apparently, here we should emphasize an important neurophysiological & neuroanatomical fact about V1-L4: It is the only neural substrate with predominantly monocular neurons in the whole cortex. We have briefly mentioned this fact in our paper: p. 6, ln. 135–148; as per your comment , we will certainly write more clearly on this point. As blind spots and afterimages are perceptually monocular, V1-L4 is both the lower and the upper boundaries for them in the whole cortex. The neurophysiological side of V1-L4 monocularity has been well-established since David Hubel and Torsent Wiesel’s Nobel prize winning work in the 1960s—and then, Dr. Jonathan Horten (who is their student) has substantially extended this fact with neuroanatomical approaches. It is this very fact allowing us to definitely conclude that V1-L4 is the neural substrate for afterimages.  Isn’t this bi-monocularity feature of V1-L4 beautiful? When one reflects upon this feature and appreciates its beauty, one can easily “hear” what Einstein used to say “Subtle is the Lord”! (see Pais, 1982).

---

> > > > ### Author Response · Authors · 2025-08-08
> > > > **Thank you very much for your comments! And here are our replies! -- CONTINUED**
> > > >
> > > > “This single observation cannot support broad claims about all afterimage mechanisms across different stimulus types, durations, and modalities.”
> > > >
> > > > As already mentioned in our rebuttal, we only claim that V1-L4 is for afterimages—that is, only about colors (including the black-gray-white dimension), not about any other visual attributes. For instance, for motion perception, there has been substantial evidence indicating that the cortical area MT is involved; and therefore, “motion aftereffect” should occur there as well—here again, please note the difference between these two terms: afterimage (pertinent only to colors) vs. aftereffect (pertinent to orientations, motions, forms, depths, etc.).
> > > >
> > > > “However, you acknowledge your descriptions need elaboration and that you're designing experiments, which reinforces my point - this work is premature for publication without empirical support.”
> > > >
> > > > By and large, whether or not a scientific discovery is premature is essentially relative. Let us look at some developments of molecular genetics during the middle of the last century. Avery et al. (1944) experiment is a good example: By now nobody will deny that their experiment is a crucial step leading to Watson and Crick’s (1953) discovery of the DNA structure and to the establishment of molecular genetics. But at the time of Avery et al. (1944), many biologists did not appreciate their result and considered it as premature (as described by Stent, 1972, p. 84). As a matter of fact, had some prominent scientists at the time treated Avery et al.’s experiment as crucial and more seriously, they might have cracked the DNA structure well before James Watson & Francis Crick accomplished their feat in 1953.
> > > >
> > > > Furthermore, science progresses step and step, one discovery building upon several preceding ones. As an example,  Watson & Crick’s (1953) discovery of the DNA structure is one of the most prominent ones in the whole history of science, but they only proposed (or say, hypothesized) a mechanism of DNA replication—the actual experiment proving this mechanism was done by Matthew Meselson and Franklin Stahl in 1958.
> > > >
> > > > “I still cannot tell how the proposed computational architecture (feedforward-STM-feedback) work from your response.”
> > > >
> > > > Let’s make an analogy here: Our proposed computational architecture can be mapped to the Production System architecture in the old, symbolic artificial intelligence framework (Russell & Norvig, 2010, p. 336). In a production system, in order to fire an if-then rule, the system’s current knowledge about the environment and about its internal states need to be put into a STM; the firing of a rule is basically a feedforward neural network (let’s say, a function mapping one level of knowledge representation to the next level), with its result being sent to the next level; and the feedback neural network is essentially the reverse of this function modifying the content of STM. As we already stated in our rebuttal, if our paper does get accepted, we will certainly clarify our computational architecture substantially more in our paper, and will include an information-flow diagram and a high-level algorithmic description.
> > > >
> > > > On the other hand, we are also looking for experts in computational domains to collaborate on fully implementing this model at an appropriate scale. A century ago, the main purpose of scientific conferences (for example, the Solvay conferences in physics and chemistry) was for exchanging scientific ideas and for face-to-face discussions. Today, as science is being conducted at a much larger scale, scientific conferences have become much larger and encompassed more functions than ever before—but we hope that NeurIPS keeps its tradition as a venue for interdisciplinary exchanges and fertilization. That is why we submitted our paper to NeurIPS in the first place.
> > > >
> > > > Have you already sensed some excitement along this line of scientific investigation? Hopefully, yes! There may be some great opportunities and challenges here for researchers coming from several backgrounds: philosophy, psychology, neuroscience, and computer science & engineering—do you or some others within the computational domains wish to jump in this endeavor? It is completely up to you!
> > > >
> > > > References
> > > >
> > > > Avery, O. T., MacLeod, C. M., & McCarty, M. (1944). Studies on the Chemical Nature of the Substance Inducing Transformation of Pneumococcal Types. Journal of Experimental Medicine. 79, 137–158
> > > >
> > > > Koch, C. (2004). The Question for Consciousness: A Neurobiological Approach. Roberts And Company Publishers.
> > > >
> > > > Pais, A. (1982). Subtle is the Lord: The Science and the Life of Albert Einstein. Oxford University Press.
> > > >
> > > > Russell, S. J., & Norvig, P. (2010). Artificial Intelligence: A Modern Approach. Prentice Hall.
> > > >
> > > > Stent, G. S. (1972). Prematurity and uniqueness in scientific discovery. Scientific American, 227, 84–93.
> > > >
> > > > Watson, J. D., & Crick, F. H. C. (1953). Molecular structure of nucleic acids: a structure for DNA. Nature, 171, 737–738.

---

### Official Review · Reviewer_Xzi1 · 2025-06-22

**Clarity:** 3
**Significance:** 2
**Originality:** 2
**Rating:** 3
**Confidence:** 3

**Summary:**

This paper challenges the conventional view on the afterimage. The authors survey the recent neuroscientific literature and propose a new perspective on the afterimage as a form of cortical short-term memory.

**Questions:**

This paper provides a new look on the STM. However, as was pointed out in the weakness, readers would be better convinced if there is a way to validate the claim made by the paper.

**Ethical Concerns:**

["NO or VERY MINOR ethics concerns only"]

**Final Justification:**

I read the authors' response and one of my points that there are lack of computational works and results to validate the proposed hypothesis is not resolved. I don't think the idea itself is bad, but I just think the Neurips is not a proper venue for this paper. I will keep the original score.

**Limitations:**

Authors talks about the limitations.

**Paper Formatting Concerns:**

No concerns

**Quality:**

2

**Strengths And Weaknesses:**

## Strength
1. Novel theoretical perspective: This paper challenges the view widely accepted by the neuroscience community. This challenge is backed by recent neuroscientific evidence.
2. Comprehensive literature review: This paper throughly surveys the related studies and draw new conclusion from them.

## Weakness
1. No original and empirical results: this paper surveys the related works, but does not have its own experimental or computational works and results. Claims of this paper is backed by the existing literature, and it limits the novelty of this paper to re-interpretation, rather than its own experimental discovery.
2. Lack of quantitative results and not computational models: As this paper talks about the computational roles of STM, it does not provie any mathematical or computational modelings for this. Therefore, it is not easy to evaluate the validity of the claim.

---

> ### Author Rebuttal · Authors · 2025-07-31
>
> Thank you very much for your careful review of our NeurIPS submission!
>
> ********************
> Weakness-1. "No original and empirical results: this paper surveys the related works, but does not have its own experimental or computational works and results. Claims of this paper is backed by the existing literature, and it limits the novelty of this paper to re-interpretation, rather than its own experimental discovery."
>
> In our opinion, we have two original contributions in our paper:
>
> 1. Correlating the La Hire phenomenon on the perceptual side with the neuroanatomical findings about the blind spot’s representation in V1-L4 by LeVay et al. (1985) in the monkey brain and by Adams et al. (2007) in the human brain. Please note that linking two previously seemingly-separate phenomena or mechanisms is also an original contribution to scientific advances. Neither LeVay et al. (1895) nor Adams et al. (2007) had made this correlation. As we are the first to suggest this correlation, we do perceive our contribution in this regard as original.
>
> The neuroanatomical findings by LeVay et al. (1985) and by Adams et al. (2007) are already functional neuroanatomy: Although their neuroanatomical staining methods were performed post-mortem shortly after the subject’s death, the observed staining patterns do reflect some persistent neuronal activities before the death. As to neuroimaging evidence, Awater et al. (2005) demonstrated a blind-spot representation in V1 in the human brain, but the current fMRI technology does not allow a fine-grain spatial resolution to locate the relevant neuronal activity to a particular layer within the cortical sheet. As to neurophysiological evidence, Komatsu et al. (2000) demonstrated that some neurons in V1-L4 as well as in V1-L6 in the monkey brain do respond to visual stimuli falling into one of the eyes’ blind spot; but we find difficult to account for why V1-L6 neurons act in such a way—that is, neurons in V1-L6 are already binocular, but the blind spot is specific to each eye and is monocular.  Between functional neuroanatomical findings on the one hand and neuroimaging & neurophysiological evidence on the other, we believe that the former are more precise and more definite—that is why we have only cited related neuroanatomical findings.
>
> References:
>
> Awater, H., Kerlin, J. R., Evans, K. K., & Tong, F. (2005). Cortical representation of space around the blind spot. Journal of Neurophysiology. 94, 3314–3324.
>
> Komatsu, H., Kinoshita, M., & Murakami, I. (2000). Neural responses in the retinotopic representation of the blind spot in the macaque V1 to stimuli for perceptual filling-in. Journal of Neuroscience, 20, 9310–9319.
>
> 2. After locating the neural substrate of visual afterimages to V1-L4, we have further proposed a computational architecture for V1 (the primary visual cortex): V1-L4 is the STM for afterimages, V1-L2&3 is the feedforward network, and V1-L4&5 is the feedback network. Please note that the cortical sheet of the human brain mainly consists of 6 layers, but L1 contains mostly neural fibers; only L2-L6 contains neurons. Our computational architecture accounts for these 5 neuron-containing layers. Of course, our current description of this proposed computational architecture needs to be elaborated. We will further address this matter in our answer to your comment Weakness-3 below.
>
> ********************
> Weakness-2. "Lack of quantitative results and not computational models: As this paper talks about the computational roles of STM, it does not provide any mathematical or computational modelings for this. Therefore, it is not easy to evaluate the validity of the claim.
>
> We will address this matter in our answer to your question below.
>
> ********************
> Question. "This paper provides a new look on the STM. However, as was pointed out in the weakness, readers would be better convinced if there is a way to validate the claim made by the paper."
>
> Thank you very much for your very insightful question here! There are two perspectives to your question:
>
> 1. The perspective from neuroscience and cognitive science
>
> As described by the British psychologist McDougall in 1901, under careful observations, some aspects of visual afterimages are quite similar to some diffusion and hysteresis phenomena observed in chemical reactions; hence, he had suggested that a chemical substance in the human eye and/or brain is responsible for afterimages—as various types of neurotransmitters were unknown at his time, McDougall referred to this candidate neurotransmitter as “X-Substances” (McDougall, 1901, p. 80).
>
> We have replicated many of McDougall’s observations about afterimages, and therefore we believe that this idea about X-Substances is worthy further consideration and exploration. In 1950s, it was discovered that the neurotransmitter “Glutamate” is the principal excitatory neurotransmitter from thalamic (that is, from the LGN) neural fibers’ terminals to cortical neurons within V1-L4 (Watkins & Jane, 2006). Correlating McDougall’s idea about X-Substances for afterimages and the present knowledge about Glutamate, we have quite naturally suggested that Glutamate’s diffusion and hysteresis is very likely the neurochemical basis for visual afterimages. One of us has presented this idea at a recent Society for Neuroscience annual meeting. We did not include this idea in our NeurIPS submission because we thought that it was too specific on the neuroscience side; as per your comment here, if our paper gets accepted, it is certainly better to be more specific with our proposed model for afterimages / STM and to include a description of this idea.
>
> We are also in collaboration with some vision scientists / psychophysicists trying to design some perceptual experiments, based on McDougall’s work (which has largely remained in obscurity, very unfortunately!), to further explore some behavioral aspects of visual afterimages and visual STM.
>
> References:
>
> McDougall, W. (1901). Some new observations ... Mind, 10, 52–97.
>
> Watkins, J. C., & Jane, D. E. (2006). The glutamate story. British Journal of Pharmacology. 147, S100–S108.
>
> 2. The computational perspective
>
> In this regard, our proposed computational architecture for the primate (including the human species) visual cortex is more relevant here—that is, each cortical area consists of three parts:  feedforward-STM-feedback. As of now, our description of this proposed computational architecture for the visual cortex is indeed not very detailed. We will certainly strengthen our paper in this regard: to describe it in more detail, and to include an information-flow diagram and a high-level algorithmic description.
>
> Our paper is more oriented on the neuroscience and cognitive science aspects; but as NeurIPS has kept its interdisciplinary tradition and as the major portion of NeurIPS attendees are from the computational domains (that is, computer scientists and engineers working on artificial neural networks, machine learning, computer vision, etc.), we believe that NeurIPS is a great and wonderful meeting venue for us to present our proposed computational architecture: Given such an opportunity, we are confident that our close interdisciplinary interactions with top talents in the computational domains would become very fruitful. We further believe that such interactions would run in two directions: (1) computer scientists would help us to further consolidate and substantiate our computational architecture as a scientific theory / model in neuroscience and cognitive science; (2) some computer scientists and/or engineers may find our computational architecture to be useful for some real-world applications.

---

> > ### Comment · Reviewer_Xzi1 · 2025-08-02
> >
> > I thank authors for the detailed response. It would be great to see the authors' collaborative works as mentioned by the response in the future Neurips conference.

---

> > > ### Author Response · Authors · 2025-08-08
> > > **Thank you very much for your review! Here we have replies to some comments!**
> > >
> > > Thank you very much for your reviews and/or comments! We have combined comments from several sources, and have put together our replies below.
> > >
> > > “While linking existing phenomena can be valuable, you provide no quantitative analysis, computational modeling, or statistical validation to support this connection.”
> > >
> > > Actually, during this year’s Vision Science Society annual meeting, we did have a demo session about the La Hire phenomenon: Over 100 attendees participated in our experiment, and we did collect some data at the time. We thought that such data (e.g., percentages of the participants seeing the phenomenon under various conditions). We thought that such detailed psychological data would be very interesting to vision researchers, but not probably so much to computer scientists—to them, knowing the very existence of this phenomenon might already be quite informative; but as per your comment, we certainly can throw in some experimental data in our paper.
> > >
> > > From your perspective, what kind of quantitative data should we furnish? We will greatly appreciate hearing any thoughtful comment in this respect!
> > >
> > > Please note that the La Hire phenomenon is not new, but it had been forgotten after WWII (due to the fact that the old research publications about this phenomenon are mainly in German)—we verified this point by both systematically looking through the relevant textbooks & journals and by asking veteran vision researchers. Just as an example, Christoph Koch is a well-known vision scientist but he explicitly denied the possibility of seeing one’s own blind spots: “Yet even if you close one eye, you still won't see a hole in your visual field” (Koch, 2004, p. 54)—This statement is wrong! Under special viewing conditions, you can indeed see one eye’s blind spot as a hole (see our paper, p. 4, Fig. 1 & lines 86–98). This is the La Hire phenomenon, and one of us re-discovered it—now, we have been resurrecting this phenomenon in the vision research community.
> > >
> > > As to predictions on the neuroscience side, when we talked to those involving fMRI on the human brain, they immediately understand what to test and what experiments to do—but unfortunately, the current fMRI technology does not have the spatial resolution to localize brain activity to a specific layer within the cortical sheet along the pia-to-white-matter axis. But fMRI technology is advancing: Siemens has already built ultrahigh resolution fMRI scanners with 11.7T magnetic field—currently, they are not safe for doing experiments with living human beings. We may see such experiments within a decade or so.
> > >
> > > Why do we think that “knowing the very existence of this phenomenon may already be quite informative”. As an example, Daniel Dennett was a well-known philosopher and his book “Consciousness Explained” contains lengthy discussions about the blind spot. Based on the assumption that it is impossible to see one’s own blind spots, he proposed his No-Representational Theory for the Blind Spot (that is, claiming that there is no neural representation for the blind spot at all in the whole brain). As we mentioned above, the La Hire phenomenon had been forgotten by the vision research community after WWII—apparently, Dennett did not know this phenomenon; had he known it, he certainly would not have proposed his incorrect theory at all. Therefore, armed with just this one bit of information (that is, knowing the existence of the La Hire phenomenon), philosophers and scientists alike will see a new perspective for further investigation—or say, for making more grounded bold guesses (as Newton once remarked: “No great scientific progress would be made without bold guesses!”).
> > >
> > > “The absence of retinal tissue at the blind spot doesn't prove V1-L4 is the afterimage substrate - it only shows cortical involvement is necessary for that specific location.”
> > >
> > > Apparently, here we should emphasize an important neurophysiological & neuroanatomical fact about V1-L4: It is the only neural substrate with predominantly monocular neurons in the whole cortex. We have briefly mentioned this fact in our paper: p. 6, ln. 135–148; as per your comment , we will certainly write more clearly on this point. As blind spots and afterimages are perceptually monocular, V1-L4 is both the lower and the upper boundaries for them in the whole cortex. The neurophysiological side of V1-L4 monocularity has been well-established since David Hubel and Torsent Wiesel’s Nobel prize winning work in the 1960s—and then, Dr. Jonathan Horten (who is their student) has substantially extended this fact with neuroanatomical approaches. It is this very fact allowing us to definitely conclude that V1-L4 is the neural substrate for afterimages.  Isn’t this bi-monocularity feature of V1-L4 beautiful? When one reflects upon this feature and appreciates its beauty, one can easily “hear” what Einstein used to say “Subtle is the Lord”! (see Pais, 1982).

---

> > > > ### Author Response · Authors · 2025-08-08
> > > > **Thank you very much for your review! Here we have replies to some comments! -- CONTINUED**
> > > >
> > > > “This single observation cannot support broad claims about all afterimage mechanisms across different stimulus types, durations, and modalities.”
> > > >
> > > > As already mentioned in our rebuttal, we only claim that V1-L4 is for afterimages—that is, only about colors (including the black-gray-white dimension), not about any other visual attributes. For instance, for motion perception, there has been substantial evidence indicating that the cortical area MT is involved; and therefore, “motion aftereffect” should occur there as well—here again, please note the difference between these two terms: afterimage (pertinent only to colors) vs. aftereffect (pertinent to orientations, motions, forms, depths, etc.).
> > > >
> > > > “However, you acknowledge your descriptions need elaboration and that you're designing experiments, which reinforces my point - this work is premature for publication without empirical support.”
> > > >
> > > > By and large, whether or not a scientific discovery is premature is essentially relative. Let us look at some developments of molecular genetics during the middle of the last century. Avery et al. (1944) experiment is a good example: By now nobody will deny that their experiment is a crucial step leading to Watson and Crick’s (1953) discovery of the DNA structure and to the establishment of molecular genetics. But at the time of Avery et al. (1944), many biologists did not appreciate their result and considered it as premature (as described by Stent, 1972, p. 84). As a matter of fact, had some prominent scientists at the time treated Avery et al.’s experiment as crucial and more seriously, they might have cracked the DNA structure well before James Watson & Francis Crick accomplished their feat in 1953.
> > > >
> > > > Furthermore, science progresses step and step, one discovery building upon several preceding ones. As an example,  Watson & Crick’s (1953) discovery of the DNA structure is one of the most prominent ones in the whole history of science, but they only proposed (or say, hypothesized) a mechanism of DNA replication—the actual experiment proving this mechanism was done by Matthew Meselson and Franklin Stahl in 1958.
> > > >
> > > > “I still cannot tell how the proposed computational architecture (feedforward-STM-feedback) work from your response.”
> > > >
> > > > Let’s make an analogy here: Our proposed computational architecture can be mapped to the Production System architecture in the old, symbolic artificial intelligence framework (Russell & Norvig, 2010, p. 336). In a production system, in order to fire an if-then rule, the system’s current knowledge about the environment and about its internal states need to be put into a STM; the firing of a rule is basically a feedforward neural network (let’s say, a function mapping one level of knowledge representation to the next level), with its result being sent to the next level; and the feedback neural network is essentially the reverse of this function modifying the content of STM. As we already stated in our rebuttal, if our paper does get accepted, we will certainly clarify our computational architecture substantially more in our paper, and will include an information-flow diagram and a high-level algorithmic description.
> > > >
> > > > On the other hand, we are also looking for experts in computational domains to collaborate on fully implementing this model at an appropriate scale. A century ago, the main purpose of scientific conferences (for example, the Solvay conferences in physics and chemistry) was for exchanging scientific ideas and for face-to-face discussions. Today, as science is being conducted at a much larger scale, scientific conferences have become much larger and encompassed more functions than ever before—but we hope that NeurIPS keeps its tradition as a venue for interdisciplinary exchanges and fertilization. That is why we submitted our paper to NeurIPS in the first place.
> > > >
> > > > Have you already sensed some excitement along this line of scientific investigation? Hopefully, yes! There may be some great opportunities and challenges here for researchers coming from several backgrounds: philosophy, psychology, neuroscience, and computer science & engineering—do you or some others within the computational domains wish to jump in this endeavor? It is completely up to you!
> > > >
> > > > References
> > > >
> > > > Avery, O. T., MacLeod, C. M., & McCarty, M. (1944). Studies on the Chemical Nature of the Substance Inducing Transformation of Pneumococcal Types. Journal of Experimental Medicine. 79, 137–158
> > > >
> > > > Koch, C. (2004). The Question for Consciousness: A Neurobiological Approach. Roberts And Company Publishers.
> > > >
> > > > Pais, A. (1982). Subtle is the Lord: The Science and the Life of Albert Einstein. Oxford University Press.
> > > >
> > > > Russell, S. J., & Norvig, P. (2010). Artificial Intelligence: A Modern Approach. Prentice Hall.
> > > >
> > > > Stent, G. S. (1972). Prematurity and uniqueness in scientific discovery. Scientific American, 227, 84–93.
> > > >
> > > > Watson, J. D., & Crick, F. H. C. (1953). Molecular structure of nucleic acids: a structure for DNA. Nature, 171, 737–738.

---

### Official Review · Reviewer_n8Py · 2025-07-01

**Clarity:** 2
**Significance:** 2
**Originality:** 3
**Rating:** 2
**Confidence:** 4

**Summary:**

The manuscript begins by presenting evidence from the literature suggesting that afterimages are primarily a cortical phenomenon rather than a retinal one. It then introduces a theoretical framework for understanding afterimages in the human visual system, built on three main pillars: (1) positive and negative afterimages share the same neural pathway, (2) afterimages can be interpreted as a form of short-term memory (STM) in the brain, and (3) computationally, STM occupies an intermediate position between feedforward and feedback networks.

**Questions:**

1. How would this STM network be implemented and how can it be systematically evaluated?

**Ethical Concerns:**

["NO or VERY MINOR ethics concerns only"]

**Final Justification:**

Just to reemphasise that, in my opinion, this is an interesting paper with great potential, but needs more details to be accessible to the NeurIPS community, and I believe the manuscript would benefit from a more detailed description of the proposed computational architecture prior to its publication in NeurIPS; therefore, I maintain my original rating.

**Limitations:**

Yes.

**Paper Formatting Concerns:**

None.

**Quality:**

3

**Strengths And Weaknesses:**

The manuscript is well written and enjoyable to read. It offers a valuable historical perspective on the phenomenon of afterimages and is well positioned within the visual science community, making a meaningful contribution to our understanding of afterimage perception. However, I have sincere doubts about NeurIPS being a suitable venue for this work—at least in its current form—since it lacks sufficient detail on how the hypothesised short-term memory circuits could be implemented and tested in relation to the afterimage phenomenon.

---

> ### Author Rebuttal · Authors · 2025-07-31
>
> Thank you very much for your careful review of our NeurIPS submission!
>
> ********************
> Weakness: “I have sincere doubts about NeurIPS being a suitable venue for this work—at least in its current form—since it lacks sufficient detail on how the hypothesized short-term memory circuits could be implemented and tested in relation to the afterimage phenomenon.”
>
> We will address this matter in our answer to your question below.
>
> ********************
> Question: “How would this STM network be implemented and how can it be systematically evaluated?”.
>
> Thank you very much for your very insightful question! There are two perspectives to this question:
>
> 1. The perspective from neuroscience and cognitive science
>
> As described by the British psychologist McDougall in 1901, under careful observations, some aspects of visual afterimages are quite similar to some diffusion and hysteresis phenomena observed in chemical reactions; hence, he had suggested that a chemical substance in the human eye and/or brain is responsible for afterimages—as various types of neurotransmitters were unknown at his time, McDougall referred to this candidate neurotransmitter as “X-Substances” (McDougall, 1901, p. 80).
>
> We have replicated many of McDougall’s observations about afterimages, and therefore we believe that this idea about X-Substances is worthy further consideration and exploration. In 1950s, it was discovered that the neurotransmitter “Glutamate” is the principal excitatory neurotransmitter from thalamic (that is, from the LGN) neural fibers’ terminals to cortical neurons within V1-L4 (Watkins & Jane, 2006). Correlating McDougall’s idea about X-Substances for afterimages and the present knowledge about Glutamate, we have quite naturally suggested that Glutamate’s diffusion and hysteresis is very likely the neurochemical basis for visual afterimages. One of us has presented this idea at a recent Society for Neuroscience annual meeting. We did not include this idea in our NeurIPS submission because we thought that it was too specific on the neuroscience side; as per your comment here, if our paper gets accepted, it is certainly better to be more specific with our proposed model for afterimages / STM and to include a description of this idea.
>
> We are also in collaboration with some vision scientists / psychophysicists trying to design some perceptual experiments, based on McDougall’s work (which has largely remained in obscurity, very unfortunately!), to further explore some behavioral aspects of visual afterimages and visual STM.
>
> References:
>
> McDougall, W. (1901). Some new observations ... Mind, 10, 52–97.
>
> Watkins, J. C., & Jane, D. E. (2006). The glutamate story. British Journal of Pharmacology. 147, S100–S108.
>
> 2. The computational perspective
>
> In this regard, our proposed computational architecture for the primate (including the human species) visual cortex is more relevant here—that is, each cortical area consists of three parts:  feedforward-STM-feedback. As of now, our description of this proposed computational architecture for the visual cortex is indeed not very detailed. We will certainly strengthen our paper in this regard: to describe it in more detail, and to include an information-flow diagram and a high-level algorithmic description.
>
> Our paper is more oriented on the neuroscience and cognitive science aspects; but as NeurIPS has kept its interdisciplinary tradition and as the major portion of NeurIPS attendees are from the computational domains (that is, computer scientists and engineers working on artificial neural networks, machine learning, computer vision, etc.), we believe that NeurIPS is a great and wonderful meeting venue for us to present our proposed computational architecture: Given such an opportunity, we are confident that our close interdisciplinary interactions with top talents in the computational domains would become very fruitful. We further believe that such interactions would run in two directions: (1) computer scientists would help us to further consolidate and substantiate our computational architecture as a scientific theory / model in neuroscience and cognitive science; (2) some computer scientists and/or engineers may find our computational architecture to be useful for some real-world applications.

---

### Official Review · Reviewer_osE5 · 2025-07-01

**Clarity:** 2
**Significance:** 2
**Originality:** 1
**Rating:** 2
**Confidence:** 4

**Summary:**

The paper explores the phenomenon of afterimages, which are visual images that persist after the original stimulus has been removed. Traditionally, it has been believed that afterimages are due to peripheral adaptation mechanisms in the retina. In this paper, the authors argue that afterimages originate in the brain, specifically in the V1, particularly in layer 4 of V1. In addition, the authors also claim that the afterimages should be conceptualized as a form of short-term memory.

**Questions:**

How would your theory account for afterimage phenomena that have been convincingly demonstrated to have retinal components, and what role, if any, do you propose for peripheral mechanisms in your framework?

**Ethical Concerns:**

["NO or VERY MINOR ethics concerns only"]

**Final Justification:**

I recommend rejecting.

**Limitations:**

Yes

**Paper Formatting Concerns:**

NIL

**Quality:**

1

**Strengths And Weaknesses:**

## Strengths
1. The paper provides a good review of the existing literature on afterimages.
2. The significance of the research question is clearly articulated.


## Weaknesses
1. My primary concern is that this paper doesn't present new experimental data or simulations to substantiate its central claims about V1-L4 localization. The arguments rely entirely on correlating existing neuroanatomical findings with phenomenological observations. For such strong theoretical claims, direct neurophysiological or neuroimaging evidence would be essential.
2. The core evidence comes from a single, specific phenomenon (i.e., seeing blind spots as afterimages). This evidential base is inadequate to support broad generalizations about all afterimage types, as different classes of afterimages (color, motion, form, duration-dependent) may involve distinct neural mechanisms that are not captured by this singular observation.
3. The proposed computational architecture (feedforward-STM-feedback) lacks detailed description, simulation and verification, with key concepts such as "short-term memory" in the context of afterimages, "variable binding," and the precise nature of the proposed neural computations remaining poorly defined.

---

> ### Author Rebuttal · Authors · 2025-07-31
>
> Thank you very much for your careful review of our NeuroIPS paper!
>
> ********************
> Weakness-1. “My primary concern …”
>
> In our opinion, we have two original contributions:
>
> 1. Correlating the La Hire phenomenon on the perceptual side with the neuroanatomical findings about the blind spot’s representation in V1-L4 by LeVay et al. (1985) in the monkey brain and by Adams et al. (2007) in the human brain. Please note that linking two previously seemingly-separate phenomena or mechanisms is also an original contribution to scientific advances. Neither LeVay et al. (1895) nor Adams et al. (2007) had made this correlation. As we are the first to suggest this correlation, we do perceive our contribution in this regard as original.
>
> The neuroanatomical findings by LeVay et al. (1985) and by Adams et al. (2007) are already functional neuroanatomy: Although their neuroanatomical staining methods were performed post-mortem shortly after the subject’s death, the observed staining patterns do reflect some persistent neuronal activities before the death. As to neuroimaging evidence, Awater et al. (2005) demonstrated a blind-spot representation in V1 in the human brain, but the current fMRI technology does not allow a fine-grain spatial resolution to locate the relevant neuronal activity to a particular layer within the cortical sheet. As to neurophysiological evidence, Komatsu et al. (2000) demonstrated that some neurons in V1-L4 as well as in V1-L6 in the monkey brain do respond to visual stimuli falling into one of the eyes’ blind spot; but we find difficult to account for why V1-L6 neurons act in such a way—that is, neurons in V1-L6 are already binocular, but the blind spot is specific to each eye and is monocular.  Between functional neuroanatomical findings on the one hand and neuroimaging & neurophysiological evidence on the other, we believe that the former are more precise and more definite—that is why we have only cited related neuroanatomical findings.
>
> References:
>
> Awater, H., Kerlin, J. R., Evans, K. K., & Tong, F. (2005). Cortical representation of space around the blind spot. Journal of Neurophysiology. 94, 3314–3324.
>
> Komatsu, H., Kinoshita, M., & Murakami, I. (2000). Neural responses in the retinotopic representation of the blind spot in the macaque V1 to stimuli for perceptual filling-in. Journal of Neuroscience, 20, 9310–9319.
>
> 2. After locating the neural substrate of visual afterimages to V1-L4, we have further proposed a computational architecture for V1 (the primary visual cortex): V1-L4 is the STM for afterimages, V1-L2&3 is the feedforward network, and V1-L4&5 is the feedback network. Please note that the cortical sheet of the human brain mainly consists of 6 layers, but L1 contains mostly neural fibers; only L2-L6 contains neurons. Our computational architecture accounts for these 5 neuron-containing layers. Of course, our current description of this proposed computational architecture needs to be elaborated. We will further address this matter in our answer to your comment Weakness-3 below.
>
> ********************
> Weakness-2. “The core evidence ...”
>
> Here we should first clarify the difference between afterimage and aftereffect. In vision science, the term “afterimage” is usually used to refer to lingering color (including the black-white dimension) sensations after the physical stimulation has been removed; on the other hand, the term “aftereffect” is usually used to refer to lingering effects of high-level visual features such as motion, orientation, and figure / form—for instance, Mather et al. (1998) discusses motion aftereffects, Webster and MacLin (1999) discusses figural aftereffects. Our paper focuses mainly on afterimages; as per your comment here, we will clarify this aspect in our paper.
>
> In normal viewing conditions, color is coupled with its original form / shape, but that is not always the case—for example, Ito (2012) demonstrated the afterimage from a circle may appear as a hexagon. In other words, colors and forms may become decoupled in afterimages. In this sense, for observing pure afterimages, a color patch without any hard border (that is, with a gradual change between the color patch and its background) is an ideal stimulus. In this regard, the afterimage from the blind spot does satisfy this condition: This spot would usually appear as a fuzzy colored patch without a hard border.
>
> Newton (1643-1727) had advocated using “crucial experiments” to evaluate scientific theories. In our opinion, the La Hire phenomenon is such a crucial experiment: There is no neural tissue devoted to the blind spot in the retina, neither in the LGN of the thalamus; only in V1-L4, there is neural tissue for the blind spod; as we can see an afterimage at the blind spot, the neural tissue in V1-L4 must be the neural substrate for the afterimage.
>
> References:
>
> Ito, H. (2012). Cortical shape adaptation transforms a circle into a hexagon: a novel afterimage illusion. Psychological Science, 23, 126–132.
>
> Mather, G., Verstraten, F., & Anstis, S. (1998). The motion aftereffect: A modern perspective. Cambridge, MA: MIT Press.
>
> Webster, M. A., & MacLin, O. H. (1999). Figural aftereffects in the perception of faces. Pyschonomic Bulletin and Review, 6, 647–653.
>
> ********************
> Weakness-3. “The proposed …”
>
> Many thanks for your very insightful comment here! There are two perspectives here:
>
> 1. The perspective from neuroscience and cognitive science
>
> As described by the British psychologist McDougall in 1901, under careful observations, some aspects of visual afterimages are quite similar to some diffusion and hysteresis phenomena observed in chemical reactions; hence, he had suggested that a chemical substance in the human eye and/or brain is responsible for afterimages—as various types of neurotransmitters were unknown at his time, McDougall referred to this candidate neurotransmitter as “X-Substances” (McDougall, 1901, p. 80).
>
> We have replicated many of McDougall’s observations about afterimages, and therefore we believe that this idea about X-Substances is worthy further consideration and exploration. In 1950s, it was discovered that the neurotransmitter “Glutamate” is the principal excitatory neurotransmitter from thalamic (that is, from the LGN) neural fibers’ terminals to cortical neurons within V1-L4 (Watkins & Jane, 2006). Correlating McDougall’s idea about X-Substances for afterimages and the present knowledge about Glutamate, we have quite naturally suggested that Glutamate’s diffusion and hysteresis is very likely the neurochemical basis for visual afterimages. One of us has presented this idea at a recent Society for Neuroscience annual meeting. We did not include this idea in our NeurIPS submission because we thought that it was too specific on the neuroscience side; as per your comment here, if our paper gets accepted, it is certainly better to be more specific with our proposed model for afterimages / STM and to include a description of this idea.
>
> We are also in collaboration with some vision scientists / psychophysicists trying to design some perceptual experiments, based on McDougall’s work (which has largely remained in obscurity, very unfortunately!), to further explore some behavioral aspects of visual afterimages and visual STM.
>
> References:
>
> McDougall, W. (1901). Some new observations ... Mind, 10, 52–97.
>
> Watkins, J. C., & Jane, D. E. (2006). The glutamate story. British Journal of Pharmacology. 147, S100–S108.
>
> 2. The computational perspective
>
> In this regard, our proposed computational architecture for the primate (including the human species) visual cortex is more relevant here—that is, each cortical area consists of three parts:  feedforward-STM-feedback. As of now, our description of this proposed computational architecture for the visual cortex is indeed not very detailed. We will certainly strengthen our paper in this regard: to describe it in more detail, and to include an information-flow diagram and a high-level algorithmic description.
>
> Our paper is more oriented on the neuroscience and cognitive science aspects; but as NeurIPS has kept its interdisciplinary tradition and as the major portion of NeurIPS attendees are from the computational domains (that is, computer scientists and engineers working on artificial neural networks, machine learning, computer vision, etc.), we believe that NeurIPS is a great and wonderful meeting venue for us to present our proposed computational architecture: Given such an opportunity, we are confident that our close interdisciplinary interactions with top talents in the computational domains would become very fruitful. We further believe that such interactions would run in two directions: (1) computer scientists would help us to further consolidate and substantiate our computational architecture as a scientific theory / model in neuroscience and cognitive science; (2) some computer scientists and/or engineers may find our computational architecture to be useful for some real-world applications.
>
> ********************
> Question. “How would your theory …?”
>
> Textbooks usually cite some seemingly-convincing evidence and arguments to show the retinal origin of afterimages—for example, “an afterimage moves with eye movement” is such an argument. But upon close scrutiny, all such arguments do not hold at all—for this eye-movement related argument, Urist (1958) (as already cited in our paper) already offered counter-arguments; but unfortunately, his paper has largely remained in obscurity. It appears that textbook writers and some researchers have a tendency to stay with a seemingly well-established notion as long as possible—until a revolutionary demonstration or finding completely overthrows the notion. As more and more psychological and neuroscience findings about afterimages are currently rapidly accumulating, we firmly believe that the notion about the retinal origin of afterimages will become completely overthrown in the next several years.

---

> > ### Comment · Reviewer_osE5 · 2025-08-02
> > **Thanks for your reply**
> >
> > Thanks for your reply. I also believe that the notion about the retinal origin of afterimages has the probability of being overthrown in the next several years.
> >
> > However, you acknowledge your descriptions need elaboration and that you're designing experiments, which reinforces my point - this work is premature for publication without empirical support. Theoretical proposals require either rigorous mathematical modeling with testable predictions or experimental validation.
> >
> > 1. Regarding response 1, your correlation between La Hire phenomenon and V1-L4 findings is purely theoretical. While linking existing phenomena can be valuable, you provide no quantitative analysis, computational modeling, or statistical validation to support this connection.
> > 2. Regarding response 2, your "crucial experiment" logic is flawed. The absence of retinal tissue at the blind spot doesn't prove V1-L4 is the afterimage substrate - it only shows cortical involvement is necessary for that specific location. This single observation cannot support broad claims about all afterimage mechanisms across different stimulus types, durations, and modalities.
> > 3. I still cannot tell how the proposed computational architecture (feedforward-STM-feedback) work from your response.
> >
> > I will keep my score.

---

> > > ### Author Response · Authors · 2025-08-08
> > > **Thank you very much for your comments! And here we have some replies!**
> > >
> > > Thank you very much for your insightful comments and for believing the same as we do: The retinal origin of afterimages may be well-overthrown in the next several years!
> > >
> > > “While linking existing phenomena can be valuable, you provide no quantitative analysis, computational modeling, or statistical validation to support this connection.”
> > >
> > > Actually, during this year’s Vision Science Society annual meeting, we did have a demo session about the La Hire phenomenon: Over 100 attendees participated in our experiment, and we did collect some data at the time. We thought that such data (e.g., percentages of the participants seeing the phenomenon under various conditions). We thought that such detailed psychological data would be very interesting to vision researchers, but not probably so much to computer scientists—to them, knowing the very existence of this phenomenon might already be quite informative; but as per your comment, we certainly can throw in some experimental data in our paper.
> > >
> > > From your perspective, what kind of quantitative data should we furnish? We will greatly appreciate hearing any thoughtful comment in this respect!
> > >
> > > Please note that the La Hire phenomenon is not new, but it had been forgotten after WWII (due to the fact that the old research publications about this phenomenon are mainly in German)—we verified this point by both systematically looking through the relevant textbooks & journals and by asking veteran vision researchers. Just as an example, Christoph Koch is a well-known vision scientist but he explicitly denied the possibility of seeing one’s own blind spots: “Yet even if you close one eye, you still won't see a hole in your visual field” (Koch, 2004, p. 54)—This statement is wrong! Under special viewing conditions, you can indeed see one eye’s blind spot as a hole (see our paper, p. 4, Fig. 1 & lines 86–98). This is the La Hire phenomenon, and one of us re-discovered it—now, we have been resurrecting this phenomenon in the vision research community.
> > >
> > > As to predictions on the neuroscience side, when we talked to those involving fMRI on the human brain, they immediately understand what to test and what experiments to do—but unfortunately, the current fMRI technology does not have the spatial resolution to localize brain activity to a specific layer within the cortical sheet along the pia-to-white-matter axis. But fMRI technology is advancing: Siemens has already built ultrahigh resolution fMRI scanners with 11.7T magnetic field—currently, they are not safe for doing experiments with living human beings. We may see such experiments within a decade or so.
> > >
> > > Why do we think that “knowing the very existence of this phenomenon may already be quite informative”. As an example, Daniel Dennett was a well-known philosopher and his book “Consciousness Explained” contains lengthy discussions about the blind spot. Based on the assumption that it is impossible to see one’s own blind spots, he proposed his No-Representational Theory for the Blind Spot (that is, claiming that there is no neural representation for the blind spot at all in the whole brain). As we mentioned above, the La Hire phenomenon had been forgotten by the vision research community after WWII—apparently, Dennett did not know this phenomenon; had he known it, he certainly would not have proposed his incorrect theory at all. Therefore, armed with just this one bit of information (that is, knowing the existence of the La Hire phenomenon), philosophers and scientists alike will see a new perspective for further investigation—or say, for making more grounded bold guesses (as Newton once remarked: “No great scientific progress would be made without bold guesses!”).
> > >
> > > “The absence of retinal tissue at the blind spot doesn't prove V1-L4 is the afterimage substrate - it only shows cortical involvement is necessary for that specific location.”
> > >
> > > Apparently, here we should emphasize an important neurophysiological & neuroanatomical fact about V1-L4: It is the only neural substrate with predominantly monocular neurons in the whole cortex. We have briefly mentioned this fact in our paper: p. 6, ln. 135–148; as per your comment , we will certainly write more clearly on this point. As blind spots and afterimages are perceptually monocular, V1-L4 is both the lower and the upper boundaries for them in the whole cortex. The neurophysiological side of V1-L4 monocularity has been well-established since David Hubel and Torsent Wiesel’s Nobel prize winning work in the 1960s—and then, Dr. Jonathan Horten (who is their student) has substantially extended this fact with neuroanatomical approaches. It is this very fact allowing us to definitely conclude that V1-L4 is the neural substrate for afterimages.  Isn’t this bi-monocularity feature of V1-L4 beautiful? When one reflects upon this feature and appreciates its beauty, one can easily “hear” what Einstein used to say “Subtle is the Lord”! (see Pais, 1982).

---

> > > > ### Author Response · Authors · 2025-08-08
> > > > **Thank you very much for your comments! And here we have some replies! -- CONTINUED**
> > > >
> > > > “This single observation cannot support broad claims about all afterimage mechanisms across different stimulus types, durations, and modalities.”
> > > >
> > > > As already mentioned in our rebuttal, we only claim that V1-L4 is for afterimages—that is, only about colors (including the black-gray-white dimension), not about any other visual attributes. For instance, for motion perception, there has been substantial evidence indicating that the cortical area MT is involved; and therefore, “motion aftereffect” should occur there as well—here again, please note the difference between these two terms: afterimage (pertinent only to colors) vs. aftereffect (pertinent to orientations, motions, forms, depths, etc.).
> > > >
> > > > “However, you acknowledge your descriptions need elaboration and that you're designing experiments, which reinforces my point - this work is premature for publication without empirical support.”
> > > >
> > > > By and large, whether or not a scientific discovery is premature is essentially relative. Let us look at some developments of molecular genetics during the middle of the last century. Avery et al. (1944) experiment is a good example: By now nobody will deny that their experiment is a crucial step leading to Watson and Crick’s (1953) discovery of the DNA structure and to the establishment of molecular genetics. But at the time of Avery et al. (1944), many biologists did not appreciate their result and considered it as premature (as described by Stent, 1972, p. 84). As a matter of fact, had some prominent scientists at the time treated Avery et al.’s experiment as crucial and more seriously, they might have cracked the DNA structure well before James Watson & Francis Crick accomplished their feat in 1953.
> > > >
> > > > Furthermore, science progresses step and step, one discovery building upon several preceding ones. As an example,  Watson & Crick’s (1953) discovery of the DNA structure is one of the most prominent ones in the whole history of science, but they only proposed (or say, hypothesized) a mechanism of DNA replication—the actual experiment proving this mechanism was done by Matthew Meselson and Franklin Stahl in 1958.
> > > >
> > > > “I still cannot tell how the proposed computational architecture (feedforward-STM-feedback) work from your response.”
> > > >
> > > > Let’s make an analogy here: Our proposed computational architecture can be mapped to the Production System architecture in the old, symbolic artificial intelligence framework (Russell & Norvig, 2010, p. 336). In a production system, in order to fire an if-then rule, the system’s current knowledge about the environment and about its internal states need to be put into a STM; the firing of a rule is basically a feedforward neural network (let’s say, a function mapping one level of knowledge representation to the next level), with its result being sent to the next level; and the feedback neural network is essentially the reverse of this function modifying the content of STM. As we already stated in our rebuttal, if our paper does get accepted, we will certainly clarify our computational architecture substantially more in our paper, and will include an information-flow diagram and a high-level algorithmic description.
> > > >
> > > > On the other hand, we are also looking for experts in computational domains to collaborate on fully implementing this model at an appropriate scale. A century ago, the main purpose of scientific conferences (for example, the Solvay conferences in physics and chemistry) was for exchanging scientific ideas and for face-to-face discussions. Today, as science is being conducted at a much larger scale, scientific conferences have become much larger and encompassed more functions than ever before—but we hope that NeurIPS keeps its tradition as a venue for interdisciplinary exchanges and fertilization. That is why we submitted our paper to NeurIPS in the first place.
> > > >
> > > > Have you already sensed some excitement along this line of scientific investigation? Hopefully, yes! There may be some great opportunities and challenges here for researchers coming from several backgrounds: philosophy, psychology, neuroscience, and computer science & engineering—do you or some others within the computational domains wish to jump in this endeavor? It is completely up to you!
> > > >
> > > > References
> > > >
> > > > Avery, O. T., MacLeod, C. M., & McCarty, M. (1944). Studies on the Chemical Nature of the Substance Inducing Transformation of Pneumococcal Types. Journal of Experimental Medicine. 79, 137–158
> > > >
> > > > Koch, C. (2004). The Question for Consciousness: A Neurobiological Approach. Roberts And Company Publishers.
> > > >
> > > > Pais, A. (1982). Subtle is the Lord: The Science and the Life of Albert Einstein. Oxford University Press.
> > > >
> > > > Russell, S. J., & Norvig, P. (2010). Artificial Intelligence: A Modern Approach. Prentice Hall.
> > > >
> > > > Stent, G. S. (1972). Prematurity and uniqueness in scientific discovery. Scientific American, 227, 84–93.
> > > >
> > > > Watson, J. D., & Crick, F. H. C. (1953). Molecular structure of nucleic acids: a structure for DNA. Nature, 171, 737–738.

---

### Note · Authors · 2025-08-15

Each human eye contains a blind spot. This fact was discovered by Edme Mariotte in 1668—with a procedure of mapping an eye’s blind spot by viewing an object disappearing into the spot. Today, this procedure is mentioned in all the related textbooks in psychology, neuroscience, and ophthalmology. This popular knowledge about the blind spot has led some prominent scientists (e.g., Francis Crick in his book “The Astonishing Hypothesis", 1995, p. 31) to assume that it is impossible to actually see one’s own blind spots in visual consciousness. Amazingly, this assumption is wrong—in fact, the phenomenon of seeing one’s own blind spots was first reported by Phillipe de La Hire in 1694; but very unfortunately, it has essentially remained in obscurity.  A few years ago, we rediscovered this phenomenon, located La Hire’s report after extensive literature search, named it the La Hire phenomenon, and have been reviving it in the vision and neuroscience research communities. This phenomenon is much more informative and significant than Mariotte's procedure, and we believe that it would be very interesting to computational researchers: It allows us to locate the neural substrate of visual STM; and this may further shed light on the computational architecture of the human visual system.

This is why we wish to introduce our research to NeurIPS; furthermore, as we are on the cognitive science and neuroscience sides, we are looking for exchanges and collaborations with computational experts.

In electromagnetism, Michael Faraday discovered the induction phenomenon in 1831 but his description was essentially phenomenological and intuitive—not mathematical at all. It was only in 1865 that James Clerk Maxwell formalized electromagnetism into a set of mathematical equations—to accomplish such a formulation, Maxwell started correspondences with Faraday in 1857 and had an in-person meeting with Faraday in 1860. Faraday and Maxwell certainly were having very different scientific backgrounds, but the exchanges between them eventually led to the establishment of classic electromagnetism. This example clearly illustrates the importance of exchanges among researchers with different backgrounds: The more exchanges, the better; the sooner, the better.  Moreover, would you like to hear a great and complete scientific story that is already accomplished by others? or rather to hear an incomplete story—and then take part in the endeavor and become a part of the story?

Thank you!

---

### Decision · Program_Chairs · 2025-09-17

**Decision:**

Reject

**Comment:**

a) The submitted paper proposes a thorough review of the visual afterimage literature. The authors claim that the afterimage effect and the blindspot are correlated. After more review of the literature they propose of computational model of V1 (by accounting for different layers).

b) The paper provide a precise description of the literature with an attempt to relate things together to question the origin of the afterimage effect.

c) The paper does not provide any new data, nor describe any mathematical model and its predictions with sufficiently enough details. Then, no quantitative results are provided. To be suitable for neurips this paper must contain a precise mathematical description of the model and at least some qualitative results illustrating the claims about the origin of afterimage.

d) The paper in its current form is not suitable for Neurips. As such, it appears as a review/opinion paper.

e) The authors have responded to the review in details but they did not provide any precise mathematical description nor qualitative and quantitative results in relation to the main claim about the afterimage effect. A lot of additional work is required for this publication to be suitable for Neurips. Yet, all reviewers recognized that the literature review is very good so we encourage the authors to advance the work in the direction of describing their model mathematically and providing quantitative and qualitative results.